# Exploring the phenomenon of intrusive mental imagery after suicide bereavement: A qualitative interview study in a British sample

**Katie Quayle**[1], **Poppy Jones**[1], **Martina Di Simplicio**[2], **Sunjeev Kamboj**[1], **Alexandra Pitman**[3,4]*

1 Research Department of Clinical, Educational and Health Psychology, UCL, London, United Kingdom,
2 Department of Brain Sciences, Imperial College London, London, United Kingdom, 3 Division of Psychiatry, UCL, London, United Kingdom, 4 Camden and Islington NHS Foundation Trust, London, United Kingdom

* a.pitman@ucl.ac.uk

## Abstract

### Introduction

Each year an estimated 48 million people are bereaved by suicide internationally. Following traumatic events, experiencing intrusive mental imagery relating to the trauma is not uncommon. This phenomenological study aimed to explore the nature, experience and impact of intrusive mental imagery after suicide bereavement.

### Methods

Semi-structured interview transcripts with 18 adults bereaved by the suicide of a close contact were analysed using thematic analysis to explore patterns and themes within the data, with particular consideration of the content of images, how people experience and relate to the imagery, and the impact that the imagery has on the bereaved.

### Results

Thematic analysis identified common characteristics in the experience of intrusive mental imagery following suicide loss, summarised under two main themes capturing: 1) the descriptive characteristics and 2) the emotional experience of intrusive mental imagery following suicide loss. The majority of participants found the experience of intrusive imagery distressing, but most also described positive aspects, including help in making sense of the death and retaining memories of the deceased.

### Conclusion

Findings inform our understanding of the distressing experience of intrusive imagery after suicide loss, also revealing perceived value in processing the death.

## Introduction

Annually nearly 700,000 people die by suicide worldwide, exposing an estimated 48 million people to suicide bereavement [1, 2]. Suicide bereavement is associated with depression, post-

**Data Availability Statement:** Anonymized transcripts are only available to researchers who are members of the UCL research team, subject to formal ethics application. Ethical approval for this

study was provided by the UCL Research Ethics Committee [reference: 16587/001; 10.02.2020]. The UCL Research Ethics Committee has restricted access to the pseudo anonymized interview transcripts stored on the UCL Data Safe Haven so that they are only accessible to members of the research team. Other researchers can apply formally to join the team to analyze data to address a specific research question. The rationale for this is that the data contain potentially identifying or sensitive information about bereaved individuals and public access risks identifying them more widely. The REC are contactable on ethics@ucl.ac. uk.

**Funding:** The authors received no specific funding for this work.

**Competing interests:** AP is a patron of the Support After Suicide Partnership; MDS has co-authored books and delivers workshops on imagery-based therapy; all authors state that they have no other conflicts of interest. This does not alter our adherence to PLOS ONE policies on sharing data and materials. All transcripts of interview recordings have been pseudonymised and de-identified, and have been securely archived in the UCL Data Safe Haven in perpetuity. Transcripts have had all identifiers removed and are labelled only by age, gender, ethnicity and time since bereavement. A condition of our ethics approval, given the sensitive nature of the data, was that transcripts will only be accessible for analysis by members of the UCL research team. Applications to join the research team as an honorary researcher will be considered on formal application setting out methodological approach and justification for the research study. The UCL Research Ethics Committee approval for this study does not permit us to make the data accessible in an open access form given the potential for deidentification based on factors in the history recounted.

traumatic stress disorder (PTSD), self-harm and suicide [2, 3]. However, an understanding of the cognitive, behavioural and biological factors that contribute to these negative outcomes remains poorly understood. This may explain the limited evidence for effective psychological interventions available for people bereaved by suicide [4–7].

Trauma-related intrusive memories are an involuntary cognitive-perceptual phenomenon characterised by distress and interference with activities of daily living [8]. Clinical experience suggests that intrusive mental imagery is a common experience after suicide loss, and suicide-bereaved people perceive a need for psychological treatment to address the trauma of suicide loss [9–11]. Intrusive imagery therefore represents a potentially important phenomenon to understand and target in psychological interventions [8, 12]. However, the experience of intrusive mental imagery has not been investigated formally in the suicide bereavement literature, and there is very limited exploration of this phenomenon in the wider bereavement literature. Existing evidence suggests that unpleasant (and pleasant) intrusive images of the death are common after bereavement by any cause, but diminish over time, with moderate-to-strong correlations of unpleasant intrusive images with depression, anxiety and complicated grief [13]. Those bereaved by violent causes of death (suicide, accident, homicide) are also more likely to experience intrusive thoughts about what the deceased may have experienced just before they died [13].

The majority of studies describing intrusive mental imagery (and interventions to address this) are situated in the context of PSTD, and whilst there are valuable insights to be gained from this clinical research there is also a need for research specific to suicide loss. PTSD is a common reaction to traumatic events, characterised by repeated and unwanted reexperiencing of the traumatic event in the form of intrusive memories and distressing dreams [14]. People bereaved by suicide are at increased risk of PTSD compared with those bereaved by other mortality causes although this finding is not consistent and only a minority of suicide-bereaved people will develop PTSD [3, 15, 16]. Intrusive imagery encapsulating themes of loss (in the context of grief or not) is also present in depression [14]. Given the critical role of intrusive mental imagery in maintaining traumatic stress and other psychological disorders, and their potential amenability to psychological treatment, it is important to understand how intrusive memories of suicide bereavement might be experienced, and how they might influence grief processing and mental health [17, 18]. This is particularly important given the possibility that such imagery contributes to the increased cognitive availability of suicide, and thereby mechanistically to suicide risk [19].

Gaining a better understanding of the nature and impact of intrusive cognitions experienced after suicide bereavement is an essential first step in clarifying whether mental imagery is an appropriate target for intervention in people bereaved by suicide experiencing intrusive imagery-related distress. In this study we aimed to investigate the phenomenon of intrusive mental imagery after suicide bereavement in a community sample by analysing data from interviews exploring individual experiences of such imagery following the suicide of a close contact.

## Methods

This study was approved by the UCL Research Ethics Committee [reference: 16587/001; 10.02.2020].

### Recruitment

We recruited a community sample of bereaved adults from the general population in the UK through an advert distributed via social media and contact lists of an umbrella group of UK

suicide bereavement support charities (Support After Suicide Partnership). Participants responding to the advert were screened for eligibility via an online questionnaire. Inclusion criteria included: adults aged 18 years and over (following bereavement in adulthood or childhood), UK residence, bereavement by suicide of a close contact (defined as a relative or friend who mattered to them and from whom they were able to obtain emotional or practical support), experience of intrusive imagery relating to that loss, and proficient English. Exclusion criteria included: recent suicide bereavement (within six months), recent suicide attempt and apparent cognitive impairment. Participants were not screened for pre-existing mental health disorders. Our risk protocol set out clear steps for managing any risk issues identified during screening or interviews. All participants gave written informed consent which included seeking the participant's general practitioner and gaining consent to contact them in the event of risks being identified. Participants were also provided with a list of support services.

## Procedure

Eligible participants were contacted by the researcher to facilitate informed consent. Purposive sampling of those eligible was used to capture different experiences of suicide bereavement (based on age, gender, ethnicity, kinship, and time since bereavement).

The interview topic guide was developed in collaboration with the Support After Suicide Partnership (see S1 Appendix) and designed to use open questioning, avoiding assuming only negative aspects of imagery. Interviews were conducted using an online video platform lasting up to 90 minutes. At the end of the interview, participants received a debrief, exploring their experience of the interview, checking wellbeing (including risk), and were sent an email containing information listing bereavement support services. One week later, participants received a follow-up email as a final debrief, giving participants the opportunity to discuss any impact of participation. Recruitment continued until saturation of themes within a diverse sample. Interviews were audio recorded and transcribed for analysis by the lead researcher to enhance familiarisation.

## Data analysis

We analysed interview transcripts using thematic analysis to identify themes within the data capturing the phenomenon of intrusive mental imagery: the content and form of the imagery, how the imagery was experienced, its emotional impact, and any relationship of the imagery to factors such as kinship to the deceased or characteristics of the relationship [20].

Initial analysis of interview transcriptions involved detailed notation of units of meaning and impression, developing initial codes. Two transcripts were independently coded by two researchers (KQ and PJ), discussing any coding differences to agree an initial coding framework. One author (KQ) then coded all remaining transcripts, collating codes into potential themes and refining themes in the iterative development of the thematic framework through discussions with the wider research team. Such discussions also involved a consideration of the differing perspectives and experiences of the analytic team to address reflexivity.

We presented our findings in line with COREQ guidelines on the reporting of qualitative research [21].

## Results

### Sample characteristics

Of those responding to the study advert, 21 potential participants were contacted for an initial eligibility screen. All were deemed suitable to participate, and 18 subsequently consented for interview. Demographic characteristics of the sample are described in Table 1.

**Table 1. Demographic characteristics of the sample (n = 18).**

|  | N (%) |
|---|---|
| **GENDER** |  |
| Male | 3 (16.7%) |
| Female | 15 (83.3%) |
| **AGE** |  |
| 18–25 years | 2 (11.1%) |
| 26–35 years | 4 (22.2%) |
| 36–45 years | 5 (27.8%) |
| 46–55 years | 3 (16.7%) |
| 56–65 years | 3 (16.7%) |
| 66–75 years | 0 (0.0%) |
| 76–85 years | 1 (5.6%) |
| **ETHNIC BACKGROUND** |  |
| White British | 14 (77.8%) |
| White Irish | 2 (11.1%) |
| Mixed Race | 1 (5.6%) |
| Other Ethnic Group | 1 (5.6%) |
| **RELATIONSHIP TO THE DECEASED** |  |
| Mother | 2 (11.1%) |
| Father | 2 (11.1%) |
| Son | 1 (5.6%) |
| Daughter | 1 (5.6%) |
| Sister | 3 (16.7%) |
| Granddaughter | 1 (5.6%) |
| Partner or Spouse | 4 (22.2%) |
| Ex-Spouse | 1 (5.6%) |
| Close Friend | 2 (11.1%) |
| Other Close Relationship | 1 (5.6%) |
| **TIME SINCE BEREAVEMENT** |  |
| Less than one year | 2 (11.1%) |
| 1–5 years | 7 (38.9%) |
| 5–10 years | 5 (27.8%) |
| 10–15 years | 2 (11.1%) |
| 15–20 years | 0 (0.0%) |
| More than 20 years | 2 (11.1%) |

## Themes identified

Thematic analysis identified two overarching themes capturing 1) the descriptive characteristics of intrusive imagery following suicide loss and 2) the cognitive and emotional responses to this. The first theme comprised three subthemes: (1.1) nature of imagery; (1.2) degree of control over imagery; and (1.3) changes in imagery over time. The second theme comprised four subthemes: (2.1) intrusive imagery is unhelpful and distressing; (2.2) impact of imagery; (2.3) ways of coping; and (2.4) positive aspects of the experience of imagery.

These themes and sub-themes are set out below with illustrative quotes. Participants' individual characteristics are shown in Table 2 for context to each quote (see S2 Appendix for distribution of themes).

**Theme 1. Descriptive characteristics of intrusive imagery.** *1.1. Nature of imagery.* **Generally, participants' descriptions of intrusive imagery identified differences in** the form and

**Table 2. Participant characteristics.**

| Participant | Gender | Age range (years) | Relationship to the deceased | Time since bereavement |
|---|---|---|---|---|
| P1 | Female | 36–45 | Ex-spouse | 3 years |
| P2 | Female | 26–35 | Close Friend | 3 years |
| P3 | Female | 26–35 | Sister | 2 years |
| P4 | Female | 36–45 | Sister | 9 years |
| P5 | Female | 26–35 | Close Friend | 2 years |
| P6 | Female | 56–65 | Spouse | 6 years |
| P7 | Female | 56–65 | Mother | 3 years |
| P8 | Female | 36–45 | Spouse | 13 years |
| P9 | Female | 76–85 | Sister | 20 years |
| P10 | Male | 46–55 | Father | 6 months |
| P11 | Female | 36–45 | Spouse | 4 years |
| P12 | Male | 46–55 | Father | 5 years |
| P13 | Female | 46–55 | Partner | 23 years |
| P14 | Female | 36–45 | Teacher | 6 months |
| P15 | Female | 56–65 | Mother | 5 years |
| P16 | Female | 18–25 | Daughter | 2 years |
| P17 | Male | 18–25 | Son | 12 years |
| P18 | Female | 26–35 | Granddaughter | 9 years |

quantity of the imagery (imagery characteristics): these defined people who experienced the same image repeatedly, those who reported a limited number of images relating to the suicide, and those who perceived multiple images relating to the suicide. For some people these patterns changed across the course of bereavement. There was substantial variation in the perceived frequency of the intrusive images, which appeared to be influenced by factors such as mood, and participants often lacked a sense of the time spent experiencing imagery. Intrusive imagery was experienced as both static images and film-like images, often first-person perspective. A key distinction appeared to be between images reported as mostly based on real memories (usually experienced by those who were present at the discovery of the deceased) and images reported as mostly based on simulating elements (experienced by both those who had and had not been present at the time of discovery). This distinction appeared to be influenced by the degree to which there was a perception of complete information about the suicide, and therefore confidence in the memories of the event. Beyond these differences in the form and quantity of the imagery, participants also described the content of the imagery as largely focused on describing the story of the deceased's suicide. While there was not one distinguishing narrative of the deceased's suicide, some key shared features emerged around the content of imagery. The following sub-themes distinguish key features of the content of the images described.

**1.1.1. Discovering the deceased.** Five participants described the discovery of the deceased as the core content of their intrusive imagery, providing images of their memories of discovering the deceased, or (if not present) how they imagined the discovery.

"*I see my son when I found him lying there dead.*" (P10).

Imagined narratives were also described as feeling very real.

"*The images in relation to the finding of him come and go, maybe because I know I'm imagining them, [they're] not necessarily real because I didn't see them with my own eyes, however they feel real.*" (P4).

**1.1.2. Suicide method as a key feature.**   For fifteen participants the method was a key feature in the intrusive imagery and was described as very graphic, even for those not present at discovery. Often, intrusive images featuring the method were located at the scene of the suicide. However sometimes imagined images featured the deceased in other contexts, for example holding items used in the suicide or appearing with the stigmata of the suicide.

"*I was imaging [him] walking down the hallway into the bedroom, but there was sort of like [items relating to the suicide method present].*" (P6).

**1.1.3. Focus on the deceased.**   Nine participants mentioned that only the deceased featured in their imagery, regardless of whether others had been present at the time.

"*Never anyone, not even the jogger that found him. . . .Nothing else, just [the deceased]. . . .But no, no nothing, nobody else at all. It all centred on him.*" (P9)

**1.1.4. Picturing the deceased as dead.**   Fifteen participants described their loved one as deceased in the imagery they experienced. For some, imagery might include aspects of events leading up to the death, but the images would always come to focus on the deceased after death.

"[The images] *weren't always right from the beginning of the day, they would be sections of it, but . . . they would always concentrate eventually on what he looked like when he died.*" (P9).

**1.1.5. Piecing together what happened at the scene of the suicide.**   Twelve participants described their images as piecing together what had happened in the last moments for the deceased.

"*one thing I really visualise is the very last moments, like I can see her now in my head [in the act of suicide].*" (P17).

For some this included considering how the deceased must have experienced those moments.

"*one thing I remember was sort of picturing [him] . . . when he was sort of thinking through what he was doing . . . what was going through his mind. And he, he must have been scared and frightened and a whole host of emotions.*" (P11).

**1.1.6. Questioning.**   Eight participants explained that their imagery reflected unanswered questions about aspects of the suicide and their search for meaning.

"*although it's a very short video, in my mind video, sort of moving image. . . it's lots of different variations of it. It's various nuances. . . . I know how it ends but I don't know which version is the right one. And I don't know . . . why it's important for me to know that.*" (P6).

*1.2. Degree of control over imagery.* **1.2.1. Lack of control.** Seventeen participants described the involuntary nature of the imagery. A lack of control over its occurrence was in itself a distressing feature of the experience, whilst imagery conjured up voluntarily was experienced as less distressing.

"*it was almost like I couldn't stop it. And I would even go back to the beginning and start again, because I got no satisfaction from it . . . it was just completely out of my control.*" (P9).

"*when. . . it's just me by myself and I'm reflecting it's different because, yeah, I think I'm more in control of it, and it's a conscious thing that I'm doing, and I think there is that healing side of it which I've never really thought about before. And I think it's different because it's not like. . . an attack, . . . it's me, myself in my personal space.*" (P18).

**1.2.2. Triggers for imagery.** Although many participants described the onset of intrusive imagery in the absence of a known trigger, eleven participants also mentioned common triggers. These included objects referencing the setting for the suicide, reminders of the deceased as an individual, or references to suicide more generally.

"*the word 'body', like 'oh the Police found a body'. If I hear anything, I see my son when I found him lying there dead.*" (P10).

*1.3. Changes in imagery over time.* For most participants the experience of intrusive imagery appeared to persist for some years following suicide bereavement. Eleven participants described a reduction in the emotional impact or frequency over time, whether spontaneously or through gaining mastery over them.

"*I think you become deadened. . . . your emotions become depleted. So that actually you become, you know, a mixture of exhausted and accepting. . . . so obviously my emotional response to visualisations and memories and events and so on, is less dramatic now . . .and more resignation I suppose. Whereas obviously in the early days it was incredibly violent and very, very difficult. . . . I couldn't breathe. I couldn't move.*" (P12).

For some, the attenuation of the distress of imagery over time appeared to be characterised by the degree to which they felt in control, expressing their ability to manage their response to the experience, reducing associated distress.

"*at the beginning it felt quite like, erm like life-debilitating because it meant that I would really struggle to go into particular bathrooms. . . . I've kind of learnt to live with the image when it comes up.*" (P16).

"*it was about being able to* [experience imagery] *but in a safe way and a controlled way. And at the beginning I had no control and it just didn't feel safe because it could come out of nowhere.*" (P6).

**Theme 2. Cognitive and emotional responses to intrusive imagery.** All participants described a range of cognitive and emotional responses to the experience of intrusive mental imagery and ways of coping. Some also described positive aspects of the experience.

*2.1. Intrusive imagery as unhelpful and distressing.* Fourteen participants described intrusive imagery as distressing or interfering with positive memories.

**2.1.1. Imagery as distressing.** Thirteen participants described their experience of intrusive imagery as distressing at various stages of the bereavement.

"*In the earlier days, you know I just wanted it to stop, you know, just very, very, very, very distressed and wanted it to stop. . . . they're still very distressing.*" (P15).

**2.1.2. Intrusive imagery interfering with positive memories of the deceased.** Seven participants described how intrusive imagery relating to the suicide could interfere with their recollection of happier memories of the deceased's life.

"*I would at some stage be grateful for, for more of the images of the positive things in [the deceased]'s life, you know, the memories. . . . I think the negative thing with these* [intrusive images] *being there, whether they're distressing or comforting, is the fact that they're taking the space . . . and preventing other images of when [the deceased] was young . . . from being there. . . . if I didn't so desperately need that connection, maybe it would be better, maybe the images would slightly fade, possibly. And allow some of those other, more positive images to come back. . .*" (P15).

Participants spoke of active attempts to recall the more pleasant memories of the deceased, but reported that this was often difficult because intrusive imagery relating to the suicide was more dominant.

"*I think I probably have more negative images than positive images. . . .I suppose one thing that I tried to kind of do . . . I would look through photos that we had from holidays, . . . his Facebook profile and all the images and posts on there. . . .I kind of force myself to review those to try and remember the happy times . . . I'd say generally the images that come to mind are associated with his death rather than his life.*" (P11).

**2.1.3. Simulations as hampering closure.** Five participants described their intrusive imagery simulating the suicide as unhelpful, preventing closure because it impeded an acceptance of not knowing what really happened.

"*I think it would just give some closure if I knew the images were real. So, if I knew exactly what she would look like, it would just stop my mind from having so many other images. I think having real images, which I know will never happen . . . I guess that would help because then it stops my, it stops your imagination.*" (P17).

*2.2. Emotional and physical impacts of intrusive imagery.* **2.2.1. *Negative emotional impacts.*** Eleven participants spoke about the negative impacts that intrusive imagery had had on their everyday lives, with the distress causing them to feel agitated, distracted, or unable to sleep, and hampering functioning.

"*I had them for months and months and months, until I was back at work, I realised that these imaginings were destroying me. I was getting quite careless at work.*" (P9).

**2.2.2. Mixed emotional responses.** Twelve participants described a range of intense, conflicting emotional responses to the intrusive imagery, which could sometimes change quickly.

"*Bereaved and just sad [tearful] and feeling like we somehow failed him. And also, really angry with why he didn't ask us to help. So, loads of conflicting stuff.*" (P7).

*it's sort of nostalgic and then sorrowful and then angry and the images just provoke rapid emotional changes like instantly and . . . there's no kind of control over that, because the triggers are so strong.*" (P4).

**2.2.3. Negative physical impacts.** Nine participants described unpleasant visceral impacts of the intrusive imagery, which arose as sudden, traumatic and uncontrollable sensations, including nausea or being frozen in whatever they were doing at the time.

*"it's so unexpected . . . It's like being punched in the stomach . . . it takes my breath away."* (P2).

*"That image can be so powerful, and the immediate kind of emotional response means that somehow or another you're, you know, quasi-paralysed. . . . It's so traumatic as to sort of freeze you on the spot. . . . I have been known to stand still in the street for no particular reason."* (P12).

*2.3. Ways of coping.* Ten participants described coping approaches that we categorized as three types of cognitive strategies they had developed.

**2.3.1. Distraction.** Five participants used distraction as a strategy to cope with the experience of intrusive imagery, but felt its effectiveness was reduced when low in mood or when prone to negative thinking styles.

*"it will depend on my mood,. . . my state of mind on a particular day. . . And if I'm not able to distract with other things then I think it will be . . . harder to pull away from the ."* (P15).

**2.3.2. Overwriting imagery.** Six participants described overwriting the intrusive imagery by attempting to swap intrusive images with more pleasant memories of the deceased, or more helpful thought processes.

*"Whenever I start imagining these things about [him] being dead in his car, and what position he was in, what did he look like and this sort of thing, I would tell myself, 'No, I'm going to think about that day when we bought those coffee tables'."* (P9).

**2.3.3. Attempts to avoid imagery.** Eight participants reported attempting to "shake off" or block out the imagery when it arose, although this was not always successful.

*". . .it depends on what the image is, usually with the ones of him you know and the actual suicide, I'm actually quite good at shaking off or having that mental block."* (P3).

*2.4. Positive aspects of the experience of intrusive imagery.* Despite the distressing experiences of intrusive imagery described, seventeen participants mentioned some positive aspects.

**2.4.1. Post-traumatic growth.** Six participants felt that intrusive imagery helped them to reflect on the enormity of a life-changing life event, and a reminder of their healing through the process of grief.

*"it reminds me of what I've been through and that I've survived . . . and that I've got this far."* (P11).

*"it's taught me resilience, it's helped me compartmentalise my thoughts and. . . it actually brings something tangible to work with, and measurable, . . . if I can feel it getting less and less frequently, I know I'm healing."* (P1).

**2.4.2. Imagery as supporting processing.** Twelve participants spoke of the role that intrusive imagery had in helping them process their suicide loss, facilitate meaning-making, and understand aspects of the suicide.

"*rather than specific images, it's more like the story . . . I'm trying to make sense of kind of what happened.*" (P11).

"*You realise actually it's your brain trying to, starting to process the whole thing that you've been through.*" (P6).

**2.4.3. Imagery as facilitating a connection with the deceased.** For seven participants imagery provided the opportunity to connect with or feel close to the deceased, providing comfort. The vivid nature of the images meant they captured a sense of having the deceased with them again.

"*it's a way to be connected to the person. . . . I still am not able really to connect, to focus on all the good memories of [her] and the life she had before. . . . But . . . because they're so strong, the images . . . it's almost as though she's still there. . . . Even though it's horrible, it's all you've got left. . . . I wonder whether the brain's trying to keep connected in that way because they're such powerful images . . . because the imagery is powerful it may be a way of connecting to her.*" (P15).

Participants expressed a reluctance for the intrusive imagery to attenuate, as this would mean losing a valued connection with the deceased.

"*[images] are . . . the last tie I have to [him], and if I didn't have those images, I wouldn't really have much to go on. . . . [the images] have kept me company for the best part of ten years and have allowed me to have kind of some kind of tie or relationship. So, I don't think I'd want to lose them. . . .it would probably feel like really letting go of [him].*" (P18).

**2.4.4. Fear of erasing the memory of the deceased.** Six participants described a reluctance to lose the imagery for fear of eroding their stock of memories of the deceased.

"*I think I might miss [the images] because it's [his] last moments. . . and anything related to him is still important. Like I would be forgetting him, or part of him.*" (P6).

**2.4.5. Imagery as important regardless of the distress.** Four participants explicitly recognised a contradiction in the value they placed on imagery despite the intense distress it could bring.

"*Should I miss [the image]. . . should I be relieved that I'd not have [it]?. . . At the moment I don't know . . . maybe I would be relieved not to have pain anymore, but I wouldn't like to not have [the image] because of the love.*" (P10).

## Patterning of themes by relationship type

We did not identify any patterning of themes by kinship group, but we did identify a pattern defined by the quality of the relationship with the deceased. Where relationships were relatively disenfranchised, intrusive imagery was experienced as more unhelpful or with less potential to provide comfort, creating a greater motivation for the bereaved to eliminate the

imagery. However, for those who described close relationships with the deceased, imagery was perceived as more important in maintaining a connection with the deceased or providing important memories of the deceased despite the associated distress. This suggests that for people bereaved by suicide, imagery can encapsulate the individual narrative of the loss, shaped particularly by the nature of the relationship with the deceased.

## Discussion

Our analysis of qualitative interview data from 18 suicide-bereaved adults described clearly the phenomenon of intrusive mental imagery following suicide loss, setting out its characteristics, as well as positive and negative aspects. The range of emotional and cognitive responses to intrusive imagery described included striking impacts on social and occupational functioning but also opportunities for meaning-making and post-traumatic growth. Individuals had developed coping strategies to deal with the negative aspects of the imagery, although with limited success. Whilst all participants found the experience upsetting, nearly all identified some positive aspects. Despite a distressing lack of control over the occurrence, frequency or intensity of the images, some participants explained that the imagery helped them make sense of the death, hold onto their memories of the deceased, and retain a sense of connection. This is an important finding given that meaning-making is viewed as a key tool in the formation of continuing bonds) [22]. Participants' awareness of the role of intrusive imagery as contributing to healing and post-traumatic growth, as coded under sub-theme 2.4.1, was also striking. Indeed, some expressed reluctance to relinquish the imagery completely for fear of losing these positive aspects.

From a phenomenological perspective our findings illustrate some parallels and contrasts with the experiences of intrusive imagery in individuals with psychological disorders, including PTSD [8, 12]. In the current study, whilst images of the suicide appeared to contain hotspots, similar to those featuring in trauma memories, often accompanied by fear, anger and sadness, they were not processed as universally threatening, as with PTSD flashbacks [14, 23, 24]. Moreover, these unpleasant emotions were sometimes accepted as an opportunity to process the trauma and thereby the price for retaining a connection with the deceased. However, for some individuals, intrusive imagery after suicide loss did hamper processing of the traumatic event and subsequent recovery, as with PTSD [14].

Similarly to intrusive imagery reported across specific psychological disorders, the occurrence of intrusive imagery following suicide loss was experienced with a distressing sense of lack of control. Participants were sometimes aware of triggers for their intrusive imagery, comparable to the triggers that are common for PTSD flashbacks [8, 12]. At other times, intrusive imagery occurred in the absence of identifiable triggers, further highlighting the uncontrollable nature of the experience, similar to that described in the context of other psychological disorders [8]. Finally, the range of cognitive and emotional responses described share similar features to responses to intrusive imagery of a trauma typically experienced in PTSD [14].

Many participants in our study identified positive aspects of their experience, contrasting with experiences of intrusive imagery described in the context of psychological disorders, where they appear largely devoid of positive aspects [8]. This finding may be understood in the context of complicated grief, where imagery is thought to relate to a yearning for the deceased and that can also be experienced as comforting [9, 13]. The positive aspects of imagery we identified in our study included the maintenance of bonds with the deceased, as consistent with the wider suicide bereavement literature describing this as a positive experience [22]. Our sub-theme entitled post-traumatic growth presents narratives that are also consistent with the quantitative literature confirming that post-traumatic growth is possible after suicide loss [25].

We noted above that imagery as a means of continuing bonds was particularly important for those with a close relationship to the deceased, and this is consistent with the reported association between secure attachment and post-traumatic growth [26]. The accounts we identified of distress and impairment coexisting with comforting aspects of imagery is consistent with those reported for imagery occurring in the context of addiction, self-harm and imagery of an individual's own suicide [12].

Overall, findings highlight similarities in the experience of intrusive mental imagery following suicide bereavement with intrusive imagery experienced in the context of PTSD, complicated grief and other psychological disorders [8]. However, certain features (e.g. specific appraisal of the positive aspects of imagery as contributing to meaning-making) may be unique to people bereaved by suicide, who face a quest to understand why their loved one chose to die, as well as coping with guilt, shame, anger, and stigma [9].

## Strengths and limitations

To our knowledge, this is the first qualitative study of the phenomenon of intrusive mental imagery experienced by people bereaved by suicide, addressing a gap in the literature and providing valuable information that can be applied to intervention development. Our use of remote video-linked interviews is likely to have enhanced disclosure and expanded the reach of the study nationally. Discussions within a multi-disciplinary research team enhanced reflexivity and improved the validity of findings. However, limitations included our sampling methods, which meant we were unable to estimate the prevalence of intrusive mental imagery following suicide loss, nor compare it to that in other bereaved groups. Despite efforts to recruit a diverse sample nationally, the sample over-represented white females, and our findings are therefore not generalizable to males or non-white individuals experiencing suicide loss. However, given cultural dimensions to the experience of intrusive mental imagery and of bereavement, we hope that this study prompts further phenomenological work sampling in specific cultural groups.

## Clinical, research and policy implications

These findings provide valuable insights for anyone supporting people bereaved by suicide by describing a distressing aspect of their grief. We also shed light on the potential mechanisms underlying the observed adverse health and social impacts of suicide bereavement, informing the development of evidence-based psychological interventions to promote recovery, in line with government suicide prevention strategy [2, 27].

Despite the distressing nature of intrusive mental imagery after suicide loss, some aspects of such imagery may benefit the bereaved individual, and interventions designed to target imagery should acknowledge such positive aspects. Approaches such as imagery rescripting, whereby individuals construct an alternative, vivid and more helpful image, which can include communicating with the deceased, may be appropriate [8]. Imagery rescripting after trauma, including after bereavement by violent death, seeks to increase control over imagery and reduce its negative emotional impact [28–30]. Such interventions could be designed to retain the positive aspects of imagery to maximise acceptability.

To inform future interventional work, further research should investigate the phenomenological experience of intrusive mental imagery in groups under-represented in this study, and how this might vary by kinship closeness and quality of the relationship with the deceased. Researching this phenomenon in a range of countries would provide an understanding of variations in the experience cross-culturally.

## Conclusions

In conclusion, this qualitative analysis of subjective experience of intrusive mental imagery following bereavement by suicide of a close contact described clear characteristics of the phenomenon and its emotional impact. Whilst intrusive mental imagery after suicide loss was universally experienced as distressing, with characteristics similar to imagery observed in common mental disorders, many described valued features. This work provides a greater understanding of the experience of suicide loss, helping professionals provide more appropriate support, and informs the development of imagery-based interventions.

## Supporting information

**S1 Appendix. Topic guide.**
(DOCX)

**S2 Appendix. Distribution of themes across 18 interview transcripts.**
(DOCX)

**S3 Appendix. Distribution of themes amongst participants.**
(DOCX)

## Acknowledgments

We acknowledge the Support After Suicide Partnership for their advice on study design and support with recruitment. We are grateful to all those who took part in interviews.

## Author Contributions

**Conceptualization:** Katie Quayle, Poppy Jones, Martina Di Simplicio, Sunjeev Kamboj, Alexandra Pitman.

**Data curation:** Katie Quayle.

**Formal analysis:** Katie Quayle, Poppy Jones.

**Investigation:** Katie Quayle.

**Methodology:** Katie Quayle, Poppy Jones, Sunjeev Kamboj, Alexandra Pitman.

**Project administration:** Katie Quayle, Poppy Jones.

**Resources:** Katie Quayle, Poppy Jones, Alexandra Pitman.

**Supervision:** Poppy Jones, Sunjeev Kamboj, Alexandra Pitman.

**Writing – original draft:** Katie Quayle.

**Writing – review & editing:** Katie Quayle, Martina Di Simplicio, Sunjeev Kamboj, Alexandra Pitman.

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
