## [Decision Letter · Decision Letter 0]

1 Sep 2022

PONE-D-22-08780Exploring the phenomenon of intrusive mental imagery after suicide bereavement: a qualitative interview study in a British samplePLOS ONE

Dear Dr. Pitman,

Thank you for submitting your manuscript to PLOS ONE. After careful consideration, we feel that it has merit but does not fully meet PLOS ONE’s publication criteria as it currently stands. Therefore, we invite you to submit a revised version of the manuscript that addresses the points raised during the review process.

The manuscript has been evaluated by three reviewers, two of whom have raised a number of concerns that need attention.

Could you please revise the manuscript to carefully address the concerns raised?

We look forward to receiving your revised manuscript.

Kind regards,

Steve Zimmerman, PhD

Associate Editor, PLOS ONE

“I have read the journal's policy and the authors of this manuscript have the following competing interests: AP is a patron of the Support After Suicide Partnership; MDS has co-authored books and delivers workshops on imagery-based therapy; all authors state that they have no other conflicts of interest.”

Reviewers' comments:

Reviewer's Responses to Questions

**Comments to the Author**

1. Is the manuscript technically sound, and do the data support the conclusions?

Reviewer #1: Partly

Reviewer #2: Yes

Reviewer #3: Yes

2. Has the statistical analysis been performed appropriately and rigorously? 

Reviewer #1: N/A

Reviewer #2: Yes

Reviewer #3: N/A

3. Have the authors made all data underlying the findings in their manuscript fully available?

Reviewer #1: No

Reviewer #2: Yes

Reviewer #3: No

4. Is the manuscript presented in an intelligible fashion and written in standard English?

Reviewer #1: Yes

Reviewer #2: Yes

Reviewer #3: Yes

5. Review Comments to the Author

Reviewer #1: Thank you for an opportunity to comment on a qualitative study exploring the phenomenon of intrusive mental imagery after suicide bereavement. The study is of interest as it explores in detail an underresearched phenomenon. I have some questions and suggestions regarding the mss.

The starting point (Introduction) is intrusive imagery in PTSD. This seems to be a quite narrow point of view to approach intrusive mental imagery in suicide bereavement. Can the authors provide information on outcomes of previous studies on intrusive mental imagery in bereavement more generally, and suicide bereavement more specifically (if such data exist)?

Further, intrusive imagery may also be related to the phenomenon of continuing bonds in suicide bereavement (eg Goodall, R., Krysinska, K., & Andriessen, K. (2022). Continuing bonds after loss by suicide: a systematic review. International journal of environmental research and public health, 19(5), 2963). This should be considered in the design of the study and discussion of results.

Further, the study sample has not been assessed for PTSD. Again, the discussion seems to be centred around this diagnosis.

Although 2.4.1 Post-traumatic growth is one of the subthemes in the analysis, it has been relatively overlooked in the the Discussion (eg Levi-Belz, Y., Krysinska, K., & Andriessen, K. (2021). “Turning personal tragedy into triumph”: A systematic review and meta-analysis of studies on posttraumatic growth among suicide-loss survivors. Psychological trauma: theory, research, practice, and policy, 13(3), 322).

Reviewer #2: Thank you for the opportunity to review this manuscript. It is well-written and provides a clear rationale for the importance of this research. A rigorous qualitative analysis has been conducted and an insightful interpretation of results provided. In my opinion, the manuscript is of a high standard and provides an important contribution to understanding the experiences and needs of people bereaved by suicide to help inform future interventions. I recommend publication.

Reviewer #3: This paper aims at investigating the phenomenon of intrusive mental imagery after suicide bereavement. Utilising qualitative approach, the authors explored individual experiences of mental imagery following the suicide of close contacts of the bereaved. Semi-structured interviews of 18 adults who experienced suicide of a close contact were conducted. Thematic analysis (TA) yield insights surrounding the characteristics of intrusive imagery, and the cognitive and emotional response to the imagery. The authors highlighted that there are some similarities with intrusive imagery experienced in PTSD, complicated grief and other psychological disorders.

The thematic analysis is clear and straightforward, but the sub-themes on the characteristics of intrusive imagery could be explored further to create a better and a more coherent link to the participants’ narrative of the deceased’s suicide. The present version is rather disjointed.

My key concern lies on the contribution of the paper. Based on the finding, the impact of intrusive mental imagery after suicide bereavement appears to be no different from other form of bereavement, and the clinical implication (from an intervention point of view) is similar. As discussed in the limitation section, the scope for generalization to other culture is very limited given the under-representation of the sample. Bereavement is a social and cultural phenomenon, and I strongly believe recognizing intrusive mental imagery cross-culturally is important.

Minor comments: Further details regarding participants such as whether they have underlying mental health conditions should be outlined. It is not clear from the exclusion criteria. I am curious whether certain pre-existing conditions might give rise to emotional complications following loss, and hence affect the interpretation of the results.

6. PLOS authors have the option to publish the peer review history of their article (what does this mean?). If published, this will include your full peer review and any attached files.

Reviewer #1: No

Reviewer #2: **Yes: **Victoria Ross

Reviewer #3: No

---

## [Author Response · Author response to Decision Letter 0]

21 Oct 2022

Reviewer #1: 

Thank you for an opportunity to comment on a qualitative study exploring the phenomenon of intrusive mental imagery after suicide bereavement. The study is of interest as it explores in detail an under-researched phenomenon. I have some questions and suggestions regarding the mss.

The starting point (Introduction) is intrusive imagery in PTSD. This seems to be a quite narrow point of view to approach intrusive mental imagery in suicide bereavement. Can the authors provide information on outcomes of previous studies on intrusive mental imagery in bereavement more generally, and suicide bereavement more specifically (if such data exist)?

We agree that the introduction cites few studies in relation to intrusive mental imagery after suicide bereavement, but we have edited this to be clear that none exist (that we are aware of) and to report the findings of the one study we were aware of in relation to intrusive mental imagery after any bereavement. We also agree that the early focus on PTSD in the Introduction disrupted the narrative arc in terms of setting out the aims of the study and implied that this study sampled those with a diagnosis of PTSD. We have edited the text (page 3-4) to make it clearer how the PTSD literature fits into this topic, explaining that this represents the majority of the literature in this area, and that whilst we can extrapolate some understanding of the phenomenon from this, we need to investigate this in a suicide-bereaved sample. We thereby justify why it is important to conduct a phenomenological study specifically in a non-clinical sample of people bereaved by suicide. 

Further, intrusive imagery may also be related to the phenomenon of continuing bonds in suicide bereavement (eg Goodall, R., Krysinska, K., & Andriessen, K. (2022). Continuing bonds after loss by suicide: a systematic review. International journal of environmental research and public health, 19(5), 2963). This should be considered in the design of the study and discussion of results.

Thank you for suggesting this additional discussion point and citation. 

Regarding the design of the study, our methodological approach was to ensure that our questioning did not only assume negative aspects of imagery, recognising the potential role of imagery as part of maintaining bonds with the deceased. Our topic guide had included questions such as 

- Do you find these images helpful or comforting in any way? 

- Is there any other way in which these images impact upon you positively or negatively?

We have now added the topic guide as an Appendix, as we felt it was important for readers to see the open questioning style, and the focus on the form of the imagery rather than the narrative of the suicide. Thus:

“The interview topic guide was developed in collaboration with the Support After Suicide Partnership (see Appendix 1) and designed to use open questioning, avoiding assuming only negative aspects of imagery.”

We have edited the discussion early on (page 22 and 23) to address this point about continuing bonds, highlighting the opportunities that positive aspects of the imagery presented participants with, in terms of making sense of the death and post-traumatic growth. We have therefore expanded our Discussion to relate our findings under theme 2.4 (positive aspects of the experience of intrusive imagery) to the literature on continuing bonds and post-traumatic growth, including this study and related ones. This relates the findings from theme 2.4 to the wider qualitative and quantitative literature in relation to suicide loss, and we believe is now much improved. 

Further, the study sample has not been assessed for PTSD. Again, the discussion seems to be centred around this diagnosis.

As we recruited a community sample, we did not use screening or diagnostic instruments to characterise our participants, but instead sought to describe the phenomenon of intrusive imagery after suicide loss, regardless of whether an individual also had the core symptoms of PTSD (including hypervigilance, nightmares, avoidance, and flashbacks). We have edited our introduction to be clear about the role of the PTSD literature in relation to what we know about intrusive imagery in the context of suicide loss. This avoids suggesting that this study focusses on a population of those with PTSD. In the discussion we relate our findings to those reported in the literature covering a range of psychological disorders, including PTSD, other anxiety disorders, depression, and addiction, and also self-harm. As the features of intrusive imagery are best described in the context PTSD, this does feature in more detail, but we do also relate our findings to those for these other psychological disorders. 

Although 2.4.1 Post-traumatic growth is one of the subthemes in the analysis, it has been relatively overlooked in the Discussion (eg Levi-Belz, Y., Krysinska, K., & Andriessen, K. (2021). “Turning personal tragedy into triumph”: A systematic review and meta-analysis of studies on posttraumatic growth among suicide-loss survivors. Psychological trauma: theory, research, practice, and policy, 13(3), 322).

We agree that although we had identified this explicitly as a sub-theme, we had neglected this in the discussion. We have now edited our discussion (page 23) to relate our findings to the quantitative summarised in Levi-Belz et al.’s meta-analysis, thus making an important link between the qualitative and the quantitative literature. 

Reviewer #2: Victoria Ross 

Thank you for the opportunity to review this manuscript. It is well-written and provides a clear rationale for the importance of this research. A rigorous qualitative analysis has been conducted and an insightful interpretation of results provided. In my opinion, the manuscript is of a high standard and provides an important contribution to understanding the experiences and needs of people bereaved by suicide to help inform future interventions. I recommend publication.

Thank you very much for your comments. 

Reviewer #3: 

This paper aims at investigating the phenomenon of intrusive mental imagery after suicide bereavement. Utilising qualitative approach, the authors explored individual experiences of mental imagery following the suicide of close contacts of the bereaved. Semi-structured interviews of 18 adults who experienced suicide of a close contact were conducted. Thematic analysis (TA) yield insights surrounding the characteristics of intrusive imagery, and the cognitive and emotional response to the imagery. The authors highlighted that there are some similarities with intrusive imagery experienced in PTSD, complicated grief and other psychological disorders.

The thematic analysis is clear and straightforward, but the sub-themes on the characteristics of intrusive imagery could be explored further to create a better and a more coherent link to the participants’ narrative of the deceased’s suicide. The present version is rather disjointed.

We were quite careful when writing up the findings of our thematic analysis not to provide too many details of the narrative relating to the deceased’s suicide, both to avoid upsetting participants and readers, and to protect participants’ anonymity. It was also because our aim was not to look for a unifying narrative emerging from the imagery of bereaved individuals (which is a different research question) but to identify the characteristics of the imagery. The themes may come across as disjointed because not all participants had a coherent narrative of the suicides and instead focussed on describing the form of the images. This is function of our topic guide, where only the first three questions probe the background to the suicide, before moving on to characteristics of the imagery. 

We had used our sub-themes within Theme 1 to capture broad features of the content of the images (discovery of the body, method used, etc). However, we have now edited the opening paragraph of theme 1 (page 9-10) to distinguish between differences in the form and quantity of the images (film-like vs static images; limited images vs multiple; etc) as an overarching feature of this theme, and specific features of the content of the images. 

To clarify this, we have added the following text to theme one:

“While there was not one distinguishing narrative of the deceased’s suicide, some key shared features emerged around the content of imagery”. 

We have also added the following text to the end of the results:

“This suggests that for people bereaved by suicide, imagery can encapsulate the individual narrative of the loss, shaped particularly by the nature of the relationship with the deceased.”

My key concern lies on the contribution of the paper. Based on the finding, the impact of intrusive mental imagery after suicide bereavement appears to be no different from other form of bereavement, and the clinical implication (from an intervention point of view) is similar. As discussed in the limitation section, the scope for generalization to other culture is very limited given the under-representation of the sample. Bereavement is a social and cultural phenomenon, and I strongly believe recognizing intrusive mental imagery cross-culturally is important.

Members of our team have previously conducted quantitative studies comparing mental health outcomes in groups bereaved by suicide and those bereaved by other causes, but our intention in this study was not to compare these groups’ experiences of intrusive imagery but to conduct a phenomenological study focussed on understanding how bereavement by suicide might influence cognitions and how intrusive imagery might be experienced in this very specific (and high risk) group. As no previous phenomenological studies have been conducted specific to suicide bereavement, we aimed to carry out the first of its kind. As we noted in our original submission, although we were able to recruit nationally, and tried hard to balance our sample across demographic characteristics, our sample was nonetheless over-represented by white females typical of studies of this kind. We agree that our findings may only have resonance to those with similar socio-demographic characteristics to those in our sample. We had acknowledged this in the limitations but have edited this section (page 24) to make this point more clearly, and to recommend that this study should prompt future studies exploring the experience of intrusive imagery in specific cultural groups. Previous research on mental imagery of suicide in clinical populations has identified both shared characteristics across cultures and more culturally-specific elements (e.g. data from the Hong Kong Mental Morbidity Survey, Ng et al. Psychiatry Res. 2016, https://pubmed.ncbi.nlm.nih.gov/27792974/), and it is likely that the same would apply to imagery in those bereaved by suicide.

Minor comments: Further details regarding participants such as whether they have underlying mental health conditions should be outlined. It is not clear from the exclusion criteria. I am curious whether certain pre-existing conditions might give rise to emotional complications following loss, and hence affect the interpretation of the results.

We have now added to our Recruitment paragraph a clarification that interview participants were not screened for pre-existing mental health disorders. Our aim had been to recruit a community sample of suicide-bereaved adults and to focus on the phenomenology of their experiences. We agree that it would be important to investigate the influence of pre-existing (and current) psychiatric symptoms on the nature and course of intrusive mental imagery after suicide loss and are planning quantitative longitudinal work to address this.

---

## [Editor Report · Decision Letter 1]

12 Apr 2023

Exploring the phenomenon of intrusive mental imagery after suicide bereavement: a qualitative interview study in a British sample

PONE-D-22-08780R1

Dear Dr. Pitman,

We’re pleased to inform you that your manuscript has been judged scientifically suitable for publication and will be formally accepted for publication once it meets all outstanding technical requirements.

Kind regards,

Barbara Dritschel, PhD

Academic Editor

PLOS ONE
---

## [Editor Report · Acceptance letter]

17 Apr 2023

PONE-D-22-08780R1 

Exploring the phenomenon of intrusive mental imagery after suicide bereavement: a qualitative interview study in a British sample 

Dear Dr. Pitman:

I'm pleased to inform you that your manuscript has been deemed suitable for publication in PLOS ONE. Congratulations! Your manuscript is now with our production department. 

Kind regards, 

on behalf of

Dr. Barbara Dritschel 

Academic Editor

PLOS ONE